# DeepOBS: A Deep Learning Optimizer Benchmark Suite

**Frank Schneider, Lukas Balles & Philipp Hennig**
University of Tübingen and Max Planck Institute for Intelligent Systems
Tübingen, Germany
`{frank.schneider,lukas.balles, ph}@tue.mpg.de`

## Abstract

There is significant past and ongoing research on optimization methods for deep learning. Yet, perhaps surprisingly, there is no generally agreed-upon protocol for the quantitative and reproducible evaluation of such optimizers. We suggest routines and benchmarks for stochastic optimization, with special focus on the unique aspects of deep learning, such as stochasticity, tunability and generalization. As the primary contribution, we present DeepOBS, a Python package of deep learning optimization benchmarks. The package addresses key challenges in the quantitative assessment of stochastic optimizers, and automates most steps of benchmarking. The library includes a wide and extensible set of ready-to-use realistic optimization problems, such as training Residual Networks for image classification on ImageNet or character-level language prediction models, as well as popular classics like MNIST and CIFAR-10. The package also provides realistic baseline results for the most popular optimizers on these test problems, ensuring a fair comparison to the competition when benchmarking new optimizers, and without having to run costly experiments. It comes with output back-ends that directly produce LaTeX code for inclusion in academic publications. It supports TensorFlow and is available open source.

## 1 Introduction

As deep learning has become mainstream, research on aspects like architectures (Graves et al., 2014; He et al., 2016; Szegedy et al., 2017; Vaswani et al., 2017; Sabour et al., 2017) and hardware (Ovtcharov et al., 2015; Chen et al., 2016; Reagen et al., 2016; Jouppi, 2016) has exploded, and helped professionalize the field. In comparison, the optimization routines used to train deep nets have arguable changed only little. Comparably simple first-order methods like SGD (Robbins & Monro, 1951), its momentum variants (Momentum) (Polyak, 1964; Nesterov, 1983) and Adam (Kingma & Ba, 2015) remain standards (Goodfellow et al., 2016; Karpathy, 2017). The low practical relevance of more advanced optimization methods is not for lack of research, though. There is a host of papers proposing new ideas for acceleration of first-order methods (Duchi et al., 2011; Tieleman & Hinton, 2012; Zeiler, 2012; Dozat, 2016; Bello et al., 2017; Loshchilov & Hutter, 2017; Reddi et al., 2018), incorporation of second-order information (Martens, 2010; Martens & Grosse, 2015; Botev et al., 2017; Zhang et al., 2017; Chen et al., 2018), and automating optimization (Schaul et al., 2013b; Mahsereci & Hennig, 2017; Rolinek & Martius, 2018), to name just a few. One problem is that these methods are algorithmically involved and difficult to reproduce by practitioners. If they are not provided in packages for popular frameworks like TensorFlow, PyTorch etc., they get little traction. Another problem, which we hope to address here, is that new optimization routines are often not convincingly compared to simpler alternatives in research papers, so practitioners are left wondering which of the many new choices is the best (and which ones even really work in the first place).

Designing an empirical protocol for deep learning optimizers is not straightforward, and the corresponding experiments can be time-consuming. This is partly due to the idiosyncrasies of the domain:

- **Generalization:** While the optimization algorithm (should) only ever see the training-set, the practitioner cares about performance of the trained model on the test set. Worse, in some important application domains, the optimizer's loss function is not the objective we ultimately care about. For instance in image classification, the real interest may be in the percentage of correctly labeled images, the accuracy. Since this 0-1 loss is infeasible in practice (Marcotte & Savard, 1992), a surrogate loss function is used instead. So which score should actually be presented in a comparison of optimizers? Train loss, because that is what the optimizer actually works on; test loss, because an over-fitting optimizer is useless, or test accuracy, because that's what the human user cares about?

- **Stochasticity:** Sub-sampling (batching) the data-set to compute estimates of the loss function and its gradient introduces stochasticity. Thus, when an optimizer is run only once on a given problem, its performance may be misleading due to random fluctuations. The same stochasticity also causes many optimization algorithms to have one or several tuning parameters (learning rates, etc.). How should an optimizer with two free parameter be compared in a fair way with one that has only one, or even no free parameters?

- **Realistic Settings, Fair Competition:** There is a widely-held belief that popular standards like MNIST and CIFAR-10 are too simplistic to serve as a realistic place-holder for a contemporary combination of large-scale data set and architecture. While this worry is not unfounded, researchers, ourselves included, have sometimes found it hard to satisfy the demands of reviewers for ever new data sets and architectures. Finding and preparing such data sets and building a reasonable architecture for them is time-consuming for researchers who want to focus on their novel algorithm. Even when this is done, one then has to not just run one's own algorithm, but also various competing baselines, like SGD, MOMENTUM, ADAM, etc. This step does not just cost time, it also poses a risk of bias, as the competition invariably receives less care than one's own method. Reviewers and readers can never be quite sure that an author has not tried a bit too much to make their own method look good, either by choosing a convenient training problem, or by neglecting to tune the competition.

To address these problems, we propose an extensible, open-source benchmark specifically for optimization methods on deep learning architectures. We make the following three contributions:

- **A protocol for benchmarking stochastic optimizers.** Section 2 discusses and recommends best practices for the evaluation of deep learning optimizers. We define three key performance indicators: final performance, speed, and tunability, and suggest means of measuring all three in practice. We provide evidence that it is necessary to show the results of multiple runs in order to get a realistic assessment. Finally, we strongly recommend reporting both loss and accuracy, for both training and test set, when demonstrating a new optimizer as there is no obvious way those four learning curves are connected in general.

- **DEEPOBS[1], a deep learning optimizer benchmark suite.** We have distilled the above ideas into an open-source python package, written in TENSORFLOW (Abadi et al., 2015), which automates most of the steps presented in section 2. The package currently provides over twenty off-the-shelf test problems across four application domains, including image classification and natural language processing, and this collection can be extended and adapted as the field makes progress. The test problems range in complexity from stochastic two dimensional functions to contemporary deep neural networks capable of delivering near state-of-the-art results on data sets such as IMAGENET. The package is easy to install in python, using the pip toolchain. It automatically downloads data sets, sets up models, and provides a back-end to automatically produce LaTeX code that can directly be included in academic publications. This automation does not just save time, it also helps researchers to create reproducible, comparable, and interpretable results.

- **Benchmark of popular optimizers** From the collection of test problems, two sets, of four simple ("small") and four more demanding ("large") problems, respectively, are selected as a core set of benchmarks. Researchers can design their algorithm in rapid iterations on the simpler set, then test on the more demanding set. We argue that this protocol saves time, while also reducing the risk of over-fitting in the algorithm design loop. The package also provides realistic baselines results for the most popular optimizers on those test problems.

---

[1]Code available at `https://github.com/fsschneider/deepobs`.

In Section 4 we report on the performance of SGD, SGD with momentum (MOMENTUM) and ADAM on the small and large benchmarks (this also demonstrates the output of the benchmark). For each optimizer we perform an exhaustive but realistic hyperparameter search. The best performing results are provided with DEEPOBS and can be used as a fair performance metric for new optimizers without the need to compute these baselines again. We invite the authors of other algorithms to add their own method to the benchmark (via a git pull-request). We hope that the benchmark will offer a common platform, allowing researchers to publicise their algorithms, giving practitioners a clear view on the state of the art, and helping the field to more rapidly make progress.

## 1.1 RELATED WORKS

To our knowledge, there is currently no commonly accepted benchmark for optimization algorithms that is well adapted to the deep learning setting. This impression is corroborated by a more or less random sample of recent research papers on deep learning optimization (Duchi et al., 2011; Zeiler, 2012; Kingma & Ba, 2015; Martens & Grosse, 2015; Dozat, 2016; Bello et al., 2017; Loshchilov & Hutter, 2017; Reddi et al., 2018), whose empirical sections follow no joint standard (beyond a popularity of the MNIST data set). There *are* a number of existing benchmarks for deep learning as such. However, they do not focus on the optimizer. Instead, they are either framework or hardware-specific, or cover deep learning as a holistic process, wrapping together architecture, hardware and training procedure, The following are among most popular ones:

**DAWNBench** The task in this challenge is to train a model for IMAGENET, CIFAR-10 or SQUAD (Rajpurkar et al., 2018) as quickly as possible to a specified validation accuracy, tuning the entire tool-chain from architecture to hardware and optimizer (Coleman et al., 2017).

**MLPerf** is another holistic benchmark similar to DAWNBench. It has two different rule sets; only the 'open' set allows a choice of optimization algorithm (MLPerf, 2018).

**Deep Learning Frameworks (Comparison)** compares runtimes of different high-level frameworks (Microsoft Machine Learning, 2018).

**DLBS** is a benchmark focused on the performance of deep learning models on various hardware systems with various software (Hewlett Packard Enterprise, 2017).

**DeepBench** tests the speed of hardware for the low-level operations of deep learning, like matrix products and convolutions (Baidu Research, 2016).

**Fathom** is another hardware-centric benchmark, which among other things assesses how computational resources are spent (Adolf et al., 2016).

**TBD** focuses on the performance of three deep learning frameworks (Zhu et al., 2018).

None of these benchmarks are good test beds for optimization research. Schaul et al. (2013a) defined unit tests for stochastic optimization. In contrast to the present work, they focus on small-scale problems like quadratic bowls and cliffs. In the context of deep learning, these problems provide unit tests, but do not give a realistic impression of an algorithm's performance in practice.

## 2 BENCHMARKING DEEP LEARNING OPTIMIZERS

This section expands the discussion from section 1 of design desiderata for a good benchmark protocol, and proposes ways to nevertheless arrive at an informative, fair, and reproducible benchmark.

### 2.1 STOCHASTICITY

The optimizer's performance in a concrete training run is noisy, due to the random sampling of mini-batches and initial parameters. There is an easy remedy, which nevertheless is not universally adhered to: Optimizers should be run on the same problem repeatedly with different random seeds, and all relevant quantities should be reported as mean and standard deviation of these samples. This allows judging the statistical significance of small performance differences between optimizers, and exposes the "variability" of performance of an optimizer on any given problem. The obvious reason why researchers are reluctant to follow this standard is that it requires substantial computational

effort. DEEPOBS alleviates this issue in two ways: It provides functionality to conveniently run multiple runs of the same setting with different seeds. More importantly, it provides stored baselines of popular optimizers, freeing computational resources to collect statistics rather than baselines.

## 2.2 CHOICE OF PERFORMANCE METRIC

Training a machine learning system is more than a pure optimization problem. The optimizers' immediate objective is *training loss*, but the users' interest is in generalization performance, as estimated on a held-out test set. It has been observed repeatedly that in deep learning, different optimizers of similar training-set performance can have surprisingly different generalization (e.g. Wilson et al. (2017)). Moreover, the loss function is regularly just a surrogate for the metric the user is ultimately interested in. In classification problems, for example, we are interested in classification accuracy, but this is infeasible to optimize directly. Thus, there are up to four relevant metrics to consider: training loss, test loss, training accuracy and test accuracy. We strongly recommend reporting all four of these to give a comprehensive assessment of a deep learning optimizer. For hyperparameter tuning, we use test accuracy or, if that is not available, test loss, as the criteria. We also use them as the performance metrics in Table 2.

For empirical plots, many authors compute train loss (or accuracy) only on mini-batches of data, since these are computed during training anyway. But these mini-batch quantities are subject to significant noise. To get a decent estimate of the training-set performance, whenever we evaluate on the test set, we also evaluate on a larger chunk of training data, which we call a *train eval* set. In addition to providing a more accurate estimate, this allows us to "switch" the architecture to evaluation mode (e.g. dropout is not used during evaluation).

## 2.3 MEASURING SPEED

Relevant in practice is not only the quality of a solution, but also the time required to reach it. A fast optimizer that finds a decent albeit imperfect solution using a fraction of other methods' resources can be very relevant in practice. Unfortunately, since learning curves have no parametric form, there is no uniquely correct way to define "time to convergence". In DEEPOBS, we take a pragmatic approach and measure the time it takes to reach an "acceptable" *convergence performance*, which is individually defined for each test problem from the baselines SGD, MOMENTUM and ADAM each with their best hyperparameter setting.

Arguably the most relevant measure of speed would be the wall-clock time to reach this convergence performance. However, wall-clock runtime has well-known drawbacks, such as dependency on hardware or weak reproducibility. So many authors report performance against gradient evaluations, since these often dominate the total computational costs. However, thiscan hide large per-iteration overhead. We recommend first measuring wall-clock time of both the new competitor and SGD on one of the small test problems for a few iterations, and computing their ratio. This computation, which can be done automatically using DEEPOBS, can be done sequentially on the same hardware. One can then report performance against the products of iterations and per-iteration cost relative to SGD.

For many first-order optimization methods, such as SGD, MOMENTUM or ADAM, the choice of hyperparameters does not affect the runtime of the algorithm. However, more evolved optimization methods, e.g. ones that dynamically estimate the Hessian, the hyperparameters can influence the runtime significantly. In those cases, it is suggested to repeat the runtime estimate for different hyperparameters.

## 2.4 HYPERPARAMETER TUNING

Almost all deep learning optimizers expose tunable hyperparameters, e.g., step sizes or averaging constants. The ease of tuning these hyperparameters is a relevant characteristic of an optimization method. How does one "fairly" compare optimizers with tunable hyperparameters?

A full analysis of the effects of an optimizer's hyperparameters on its performance and speed is tedious, especially since they often interact. Even a simpler sensitivity analysis requires a large number of optimization runs, which are infeasible for most users. Such analyses also do not take into

account if hyperparameters have default values that work for almost all optimization problems and therefore require no tuning in general. Instead we recommend that authors find and report the best-performing hyperparameters for each test problem. Since DEEPOBS covers multiple test problems, the spread of these best choices gives a good impression of the required tuning. Additionally, we suggest reporting the relative performance of the hyperparameter settings used during this tuning process (Figure 3 shows an example). Doing so yields a characterization of tunability without additional computations.

For the baselines presented in this paper, we chose a simple log-grid search to tune the learning rate. While this is certainly not an optimal tuning method, and more sophisticated methods exists (e.g. (Bergstra & Bengio, 2012), (Snoek et al., 2012)), it is nevertheless used often in practice and reveals interesting properties about the optimizers and their tunability. Other tuning methods can be used with DEEPOBS however, this would require recomputing the baselines as well.

DEEPOBS supports authors in adhering to good scientific practice by removing various moral hazards. The baseline results for popular optimizers (whose hyperparameters have been tuned by us or, in the future, the very authors of the competing methods) avoid "starving" the competition of attention. When using different hyperparameter tuning methods, it is necessary to allocate the same computational budget for all methods in particular when comparing optimization methods of varying number of hyperparameters.

The fixed set of test problems provided by the benchmark makes it impossible to (knowingly or subconsciously) cherry-pick problems tuned to a new method. And finally, the fact that the benchmark spreads over multiple such problem sets constitutes a mild but natural barrier to "overfit" the optimizer method to established data sets and architectures (like MNIST).

## 3 BENCHMARK SUITE OVERVIEW

DEEPOBS provides the full stack required for rapid, reliable, and reproducible benchmarking of deep learning optimizers. At the lowest level, a **data loading** (§3.1) module automatically loads and pre-processes data sets downloaded from the net. These are combined with a list of **models** (§3.2) to define test problems. At the core of the library, **runners** (§3.3) take care of the actual training, and log a multitude of statistics, e.g., training loss or test accuracy. **Baselines** (§3.4) are provided for a collection of competitors. They currently include the popular choices SGD (raw, and with MOMENTUM) and ADAM, but we invite authors of other methods to contribute their own. The **visualization** (§3.6) script maps the results to LATEX output.

Future releases of DEEPOBS will include a version number that follows the pattern MAJOR.MINOR.PATCH, where MAJOR versions will differ in the selection of the benchmark sets, MINOR versions signify changes that could affect the results. PATCHES will not affect the benchmark results. All results obtained with the same MAJOR.MINOR version of DEEPOBS will be directly comparable, all results with the same MAJOR version will compare results on the same problems.

We now give a brief overview of the functionality; the full documentation can be found online.[2]

Figure 1: Illustration of the different steps implemented in the DEEPOBS package and their outputs. The color of each block highlights the way a user mostly interacts with this part. Blocks in ● signify classes, those in ● are used via command line scripts. ● signals data packaged with DEEPOBS and ● denotes parts provided through template scripts.

___________________________

[2] https://deepobs.readthedocs.io/

## 3.1 Data loading

DEEPOBS can automatically download and pre-process all necessary data sets.[3] Excluding IMA-GENET, the downloaded data sets require less than one GB of disk space.

The DEEPOBS data loading module then performs all necessary processing of the data sets to return inputs and outputs for the deep learning model (e.g. images and labels for image classification). This processing includes splitting, shuffling, batching and data augmentation. The data loading module can also be used to build new deep learning models that are not (yet) part of DEEPOBS.

## 3.2 Models

Together, data set and model define a loss function and thus an optimization problem. Table 1 provides an overview of the data sets and models included in DEEPOBS. We selected problems for diversity of task as well as the difficulty of the optimization problem itself. The list includes popular image classification models on data sets like MNIST, CIFAR-10 or IMAGENET, but also models for natural language processing and generative models. Additionally, three two-dimensional problems and an ill-conditioned quadratic problem are included. These simple tests can be used as illustrative toy problems to highlight properties of an algorithm and perform sanity-checks. Over time, we plan to expand this list when hardware and research progress renders small problems out of date, and introduces new research directions and more challenging problems.

Table 1: Overview of the test problems included in DEEPOBS with their properties showing if the test problem includes convolutional layers (*Conv*), recurrent neural network cells (*RNN*), dropout layers (*Drop*), batch normalization layers (*BN*) or weight decay (*WD*). The first column highlights the machine learning task that the model performs, i.e. image classification ● , generative model ● , natural language processing ● or problems where the loss function is given explicitly ● . Test problems marked in ▮ and ▮ are part of the small and large benchmark set, respectively.

| | Data set | Model | Description | Conv | RNN | Drop | BN | WD |
|---|---|---|---|---|---|---|---|---|
| ● | | Noisy Beale | *Noisy version of the Beale function* | | | | | |
| ● | 2D | Noisy Branin | *Noisy version of the Branin function (Branin, 1972)* | | | | | |
| ● | | Noisy Rosenbrock | *Noisy version of the Rosenbrock function (Rosenbrock, 1960)* | | | | | |
| ● | Quadratic | Deep | 100*-dimensional ill-conditioned noisy quadratic (Chaudhari et al., 2017)* | | | | | |
| ● | | Log. Regr. | *Logistic regression* | | | | | |
| ● | MNIST | MLP | *Four layer fully-connected network* | | | | | |
| ● | (Lecun et al., 1998) | 2c2d | *Two conv. and two fully-connected layers* | ✓ | | | | |
| ● | | VAE | *Variational Autoencoder* | ✓ | | ✓ | | |
| ● | FASHION | Log. Regr. | *Logistic regression* | | | | | |
| ● | MNIST | MLP | *Four layer fully-connected network* | | | | | |
| ● | (Xiao et al., 2017) | 2c2d | *Two conv. and two fully-connected layers* | ✓ | | | | |
| ● | | VAE | *Variational Autoencoder* | ✓ | | ✓ | | |
| ● | CIFAR-10 | 3c3d | *Three conv. and three fully-connected layers* | ✓ | | | | ✓ |
| ● | (Krizhevsky & Hinton, 2009) | VGG 16 | *Adapted version of VGG 16 (Simonyan & Zisserman, 2014)* | ✓ | | ✓ | | ✓ |
| ● | | VGG 19 | *Adapted version of VGG 19* | ✓ | | ✓ | | ✓ |
| ● | | 3c3d | *Three conv. and three fully-connected layers* | ✓ | | | | ✓ |
| ● | CIFAR-100 | VGG 16 | *Adapted version of VGG 16* | ✓ | | ✓ | | ✓ |
| ● | (Krizhevsky & Hinton, 2009) | VGG 19 | *Adapted version of VGG 19* | ✓ | | ✓ | | ✓ |
| ● | | All-CNN-C | *The all convolutional net from Springenberg et al. (2015)* | ✓ | | ✓ | | ✓ |
| ● | | Wide ResNet-40-4 | *Wide Residual Network (Zagoruyko & Komodakis, 2016)* | ✓ | | | ✓ | ✓ |
| ● | SVHN | 3c3d | *Three conv. and three fully-connected layers* | ✓ | | | | ✓ |
| ● | (Netzer et al., 2011) | Wide ResNet-16-4 | *Wide Residual Network* | ✓ | | | ✓ | ✓ |
| ● | IMAGENET | VGG 16 | *VGG 16* | ✓ | | ✓ | | ✓ |
| ● | (Deng et al., 2009) | VGG 19 | *VGG 19* | ✓ | | ✓ | | ✓ |
| ● | | Inception-v3 | *Inception-v3 network as described by Szegedy et al. (2016)* | ✓ | | ✓ | ✓ | ✓ |
| ● | Tolstoi | CharRNN | *Recurrent Neural Network for character-level language modeling* | | ✓ | ✓ | | |

## 3.3 Runners

The runners of the DEEPOBS package handle training and the logging of statistics measuring the optimizers performance. For optimizers following the standard TensorFlow optimizer API it is enough to provide the runners with a list of the optimizer's hyperparameters. We provide a template for this, as well as an example of including a more sophisticated optimizer that can't be described as a subclass of the TensorFlow optimizer API.

---

[3]At the moment, IMAGENET is not part of this automatic procedure, since IMAGENET requires registration to download the data set, and is comparably large, thus impractical for many users.

### 3.4 BASELINES

DEEPOBS also provides realistic baselines results for, currently, the three most popular optimizers in deep learning, SGD, MOMENTUM, and ADAM. These allow comparing a newly developed algorithm to the competition without computational overhead, and without risk of conscious or unconscious bias against the competition. Section 4 describes how these baselines were constructed and discusses their performance.

Baselines for further optimizers will be added when authors provide the optimizer's code, assuming the method perform competitively. Currently, baselines are available for all test problems in the small and large benchmark set; we plan to provide baselines for the full set of models in the near future.

### 3.5 ESTIMATE RUNTIME

DEEPOBS provides an option to quickly estimate the runtime overhead of a new optimization method compared to SGD. It measures the ratio of wall-clock time between the new optimizer and SGD. By default this ratio is measured on five runs each, for three epochs, on a fully connected network on MNIST. However, this can be adapted to a setting which fairly evaluates the new optimizer, as some optimizers might have a high initial cost that amortizes over many epochs.

### 3.6 VISUALIZATIONS

The DEEPOBS visualization module reduces the overhead for the preparation of results, and simultaneously standardizes the presentation, making it possible to include a comparably large amount of information in limited space. The module produces `.tex` files with `pgfplots`-code for all learning curves for the proposed optimizer as well as the most relevant baselines (section 4 includes an example of this output).

## 4 INSIGHTS FROM THE BASELINES

For the baseline results provided with DEEPOBS, we evaluate three popular deep learning optimizers (SGD, MOMENTUM and ADAM) on the eight test problems that are part of the small (problems P1 to P4) and large (problems P5 to P8) benchmark set (cf. Table 1 or the appendix). The experiments were done with version 1.1.0 of DEEPOBS. All experiments used $0.99$ for the MOMENTUM parameter and default parameters for ADAM ($\beta_1 = 0.9$, $\beta_2 = 0.999$, $\epsilon = 10^{-8}$). The learning rate $\alpha$ was tuned for each optimizer and test problem individually, by evaluating on a logarithmic grid from $\alpha_{\min} = 10^{-5}$ to $\alpha_{\max} = 10^2$ with 36 samples. Once the best learning rate has been determined, we run those settings ten times with different random seeds. While we are using a log grid search, researchers are free to use any other hyperparameter tuning method, however this would require re-running the baselines as well.

Figure 2 shows the learning curves of the eight problems in the small and large benchmark set. Table 2 summarizes the results from both benchmark sets. We focus on three main observations, which corroborate widely-held beliefs and support the case for an extensive and standardized benchmark.

**There is no optimal optimizer for all test problems.** While ADAM compares favorably on most test problems, in some cases the other optimizers are considerably better. This is most notable on CIFAR-100, where MOMENTUM is significantly better then the other two.

**The connection between the four learning metrics is non-trivial.** Looking at P6 and P7 we note that the optimizers rank differently on train vs. test loss. However, there is no optimizerthat universally generalizes better than the others; the generalization performance is evidently problem-dependent. The same holds for the generalization from loss to accuracy (e.g. P3 or P6).

**ADAM is *somewhat* easier to tune.** Between the eight test problems, the optimal learning rate for each optimizer varies significantly. Figure 3 shows the final performance against learning rate for each of the eight test problems. There is no significant difference between the three optimizers in terms of their learning rate sensitivity. However, in most cases, the order of magnitude of the optimal learning rate for ADAM is in the order of $10^{-4}$ and $10^{-3}$ (with the exception of P1), while for SGD and MOMENTUM this spread is slightly larger.

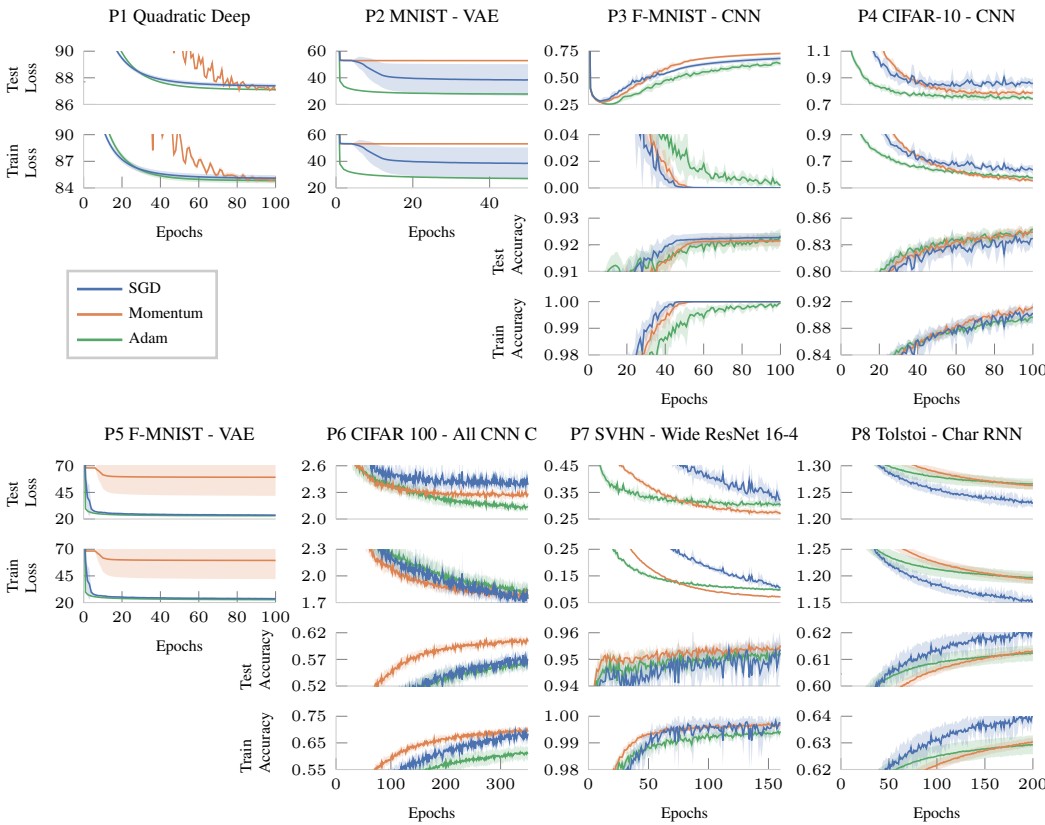

Figure 2: Learning curves for all eight test problems showing the performance of SGD, MOMENTUM, and ADAM produced with DEEPOBS.

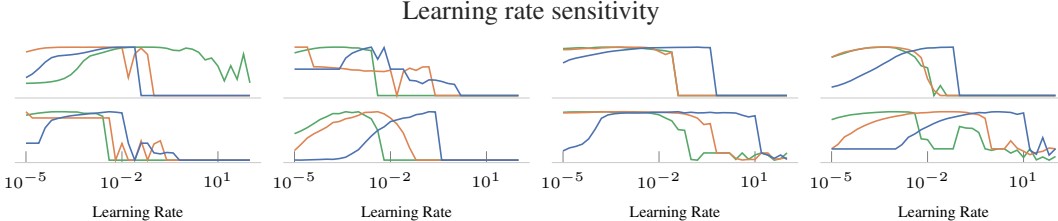

Figure 3: Relative performance against learning rate for each test problem and optimizer. Top row shows test problems P1 to P4, bottom row the test problems P5 to P8. The optimizers are represented in the same color as in Figure 2, where ● represents SGD, ● represents MOMENTUM, and ● the ADAM optimizer.

## 5 CONCLUSION

Deep learning continues to pose a challenging domain for optimization algorithms. Aspects like stochasticity and generalization make it challenging to benchmark optimization algorithms against each other. We have discussed best practices for experimental protocols, and presented the DEEPOBS package, which provide an open-source implementation of these standards. We hope that DEEPOBS can help researchers working on optimization for deep learning to build better algorithms, by simultaneously making the empirical evaluation simpler, yet also more reproducible and fair. By providing a common ground for methods to be compared on, we aim to speed up the development of deep-learning optimizers, and aid practitioners in their decision for an algorithm.

Table 2: DEEPOBS benchmark for the baseline optimizers, showing the performance, speed and tuneability measures for SGD, MOMENTUM and ADAM on all eight test problems. The performance is measured using the test accuracy in percent (when available, otherwise the test loss) and the speed using the number of iterations to reach the convergence performance. All numbers are averaged over ten runs with the same hyperparameter settings. The tuneability row indicates the best performing set of hyperparameters per test problem.

| Test Problem | | SGD | Momentum | Adam |
|---|---|---|---|---|
| P1 **Quadratic Deep** | Performance | 87.40 | 87.05 | 87.11 |
| | Speed | 51.1 | 70.5 | 39.9 |
| | Tuneability | $\alpha$: 1.58e-02 | $\alpha$: 2.51e-03 $\mu$: 0.99 | $\alpha$: 3.98e-02 $\beta_1$: 0.9 $\beta_2$: 0.999 $\epsilon$: 1e-08 |
| P2 **MNIST VAE** | Performance | 38.46 | 52.93 | 27.83 |
| | Speed | 1.0 | 1.0 | 1.0 |
| | Tuneability | $\alpha$: 3.98e-03 | $\alpha$: 2.51e-05 $\mu$: 0.99 | $\alpha$: 1.58e-04 $\beta_1$: 0.9 $\beta_2$: 0.999 $\epsilon$: 1e-08 |
| P3 **F-MNIST CNN** | Performance | 92.27 % | 92.14 % | 92.34 % |
| | Speed | 40.6 | 59.1 | 40.1 |
| | Tuneability | $\alpha$: 1.58e-01 | $\alpha$: 2.51e-03 $\mu$: 0.99 | $\alpha$: 2.51e-04 $\beta_1$: 0.9 $\beta_2$: 0.999 $\epsilon$: 1e-08 |
| P4 **CIFAR-10 CNN** | Performance | 83.71 % | 84.41 % | 84.75 % |
| | Speed | 42.5 | 40.7 | 36.0 |
| | Tuneability | $\alpha$: 6.31e-02 | $\alpha$: 3.98e-04 $\mu$: 0.99 | $\alpha$: 3.98e-04 $\beta_1$: 0.9 $\beta_2$: 0.999 $\epsilon$: 1e-08 |

| Test Problem | | SGD | Momentum | Adam |
|---|---|---|---|---|
| P5 **F-MNIST VAE** | Performance | 23.80 | 59.23 | 23.07 |
| | Speed | 1.0 | 1.0 | 1.0 |
| | Tuneability | $\alpha$: 3.98e-03 | $\alpha$: 1.00e-05 $\mu$: 0.99 | $\alpha$: 1.58e-04 $\beta_1$: 0.9 $\beta_2$: 0.999 $\epsilon$: 1e-08 |
| P6 **CIFAR-100 All CNN C** | Performance | 57.06 % | 60.33 % | 56.15 % |
| | Speed | 128.7 | 72.8 | 152.6 |
| | Tuneability | $\alpha$: 1.58e-01 | $\alpha$: 3.98e-03 $\mu$: 0.99 | $\alpha$: 1.00e-03 $\beta_1$: 0.9 $\beta_2$: 0.999 $\epsilon$: 1e-08 |
| P7 **SVHN Wide ResNet** | Performance | 95.37 % | 95.53 % | 95.25 % |
| | Speed | 28.3 | 10.8 | 12.1 |
| | Tuneability | $\alpha$: 2.51e-02 | $\alpha$: 6.31e-04 $\mu$: 0.99 | $\alpha$: 1.58e-04 $\beta_1$: 0.9 $\beta_2$: 0.999 $\epsilon$: 1e-08 |
| P8 **TOLSTOI Char RNN** | Performance | 62.07 % | 61.30 % | 61.23 % |
| | Speed | 47.7 | 88.0 | 62.8 |
| | Tuneability | $\alpha$: 1.58e+00 | $\alpha$: 3.98e-02 $\mu$: 0.99 | $\alpha$: 2.51e-03 $\beta_1$: 0.9 $\beta_2$: 0.999 $\epsilon$: 1e-08 |

## ACKNOWLEDGMENTS

The authors thank the International Max Planck Research School for Intelligent Systems (IMPRS-IS) for supporting Frank Schneider and Lukas Balles. Lukas Balles and Philipp Hennig gratefully acknowledge support by the ERC action StG 757275 / PANAMA.

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

## A  EXPERIMENTAL SETUP

We describe the in total eight test problems that are part of the small and the large benchmark set.

P1 **Quadratic Deep:** A 100 dimensional stochastic quadratic loss function. $90\%$ of the eigenvalues are drawn from $[0, 1]$, and $10\%$ from $[30, 60]$ creating an ill-conditioned problem with a structured eigenspectrum similar to the one reported by Chaudhari et al. (2017). We train with a batch size of 128 for 100 epochs.

P2 **MNIST — VAE:** A variational autoencoder (Kingma & Welling, 2014) with three convolutional and three deconvolutional layers with dropout layers and a latent space of size 8 on the MNIST data set. Trained with a batch size of 64 for 50 epochs.

P3 **FASHION-MNIST — CNN:** A vanilla convolutional network with two convolutional and two fully connected layers for image classification on the FASHION-MNIST data set. Trained with a batch size of 128 for 100 epochs.

P4 **CIFAR-10 — CNN:** A slightly larger convolutional network with three convolutional and three fully connected layers on CIFAR-10. Trained with a batch size of 128 for 100 epochs.

P5 **FASHION-MNIST — VAE:** A variational autoencoder with three convolutional and three deconvolutional layers with dropout layers and a latent space of size 8 on the FASHION-MNIST data set. Trained for 100 epochs with a batch size of 64.

P6 **CIFAR-100 — All-CNN-C:** The all convolutional network All-CNN-C from Springenberg et al. (2015) for image classification on the CIFAR-100 data set. Trained with a batch size of 256 for 350 epochs.

P7 **STREET VIEW HOUSE NUMBERS — Wide ResNet-16-4:** The wide residual network WRN-16-4 architecture of Zagoruyko & Komodakis (2016) on the STREET VIEW HOUSE NUMBERS data set for image classification. Trained with a batch size of 128 for 160 epochs.

P8 **TOLSTOI — CharRNN:** A two-layer LSTM (Hochreiter & Schmidhuber, 1997) with 128 units per LSTM cell for character-level language modeling on TOLSTOI's WAR AND PEACE. Trained with a sequence length of 50 and batch size of 50 for 200 epochs.

