# OpenReview forum: "DeepOBS: A Deep Learning Optimizer Benchmark Suite"
_ICLR.cc/2019/Conference_

### Official Review · AnonReviewer1 · 2018-11-02
**An important and useful tool for the field.**

**Rating:** 7
**Confidence:** 4

**Review:**

The authors propose a benchmark for optimization algorithms specific to deep learning called DeepOBS. They provide code to evaluate an optimizer against a suite of standard tasks in deep learning, and provide well tuned baselines for a comparison. The authors discuss important considerations when comparing optimizers, including how to measure speed and tunability of an optimizer, what metric(s) to compare against, and how to deal with stochasticity.

A clear, standardized optimization benchmark suite would be very valuable for the field. As the others clearly state in the introduction, there have been many proposed optimization algorithms, but it is hard to compare many of these due to differences in how the optimizers were evaluated in the original papers. In general, people have different requirements for what the expect from an optimizer. However, this paper does a good job of discussing most of the factors that people should consider when choosing or comparing optimizers. Providing a set of well tuned baselines would save people a lot of time in making comparisons with a new optimizer, as well as providing a canonical set of tasks to evaluate against. I particularly appreciated the breadth and diversity of the included tasks.

I am a little worried that people will still find minor quibbles with particular choices or tasks in this suite, and therefore continue to use bespoke comparisons, but I think this benchmark would be a valuable resource for the community.

Some minor comments:
- In section 2.3, there is a recommendation for how to estimate per-iteration cost. I would mention in this section that this procedure is automated and part of the benchmark suite.
- I wanted to see how the baselines performed on all of the tasks in the suite (not just on the 8 tasks in the benchmark sets). Perhaps those figures could be included in an appendix.
- The authors might want to consider including an automated way of generating performance profiles (https://arxiv.org/abs/cs/0102001) across tasks as part of DeepOBS, as a way of getting a sense of how optimizers performed generally across all tasks.

---

> ### Author Response · Authors · 2018-11-07
> **Response to the Comments of Reviewer 1**
>
> Dear Reviewer 1.
>
> thank you very much for your positive review.
>
> We want to address the minor comments you raised.
> -  We added a remark in section 2.3 regarding the automated estimation of per-iteration cost in DeepOBS.
> - With the current setup, computing the baseline performances on all 26 test problems would require more than 3500 runs. As these test problems also include the ImageNet data set, this could take quite a while. We therefore doubt, whether we could finish this in time for ICLR. However, we will add these results to the DeepOBS package as soon as they are finished so that the software package has baselines performances for all test problems.
> - Thank you for the reference. We will look into performance profiles to see how we can use them.

---

### Official Review · AnonReviewer3 · 2018-11-02
**Good initiative at its beginning stage**

**Rating:** 6
**Confidence:** 4

**Review:**

As the paper claims there is no common accept system for benchmarking deep learning optimizer. It is also hard to repeat others' results. The paper describes a benchmarking framework for deep learning optimizer. It proposes three performance indicators, and includes 20 test problems and a core set of benchmarks.

Pro:
1) It is a very relevant project. There is a need for unified benchmarking framework. In traditional optimization field, benchmarking is well studied and architectured. See an example at http://plato.asu.edu/bench.html
2) The system is at its early stage, but its design seems complete
3) The paper shows some performance of vanilla SGD, momentum, and Adam

Con:
1) It will take tremendous efforts to convince others to join the party and contribute
2) It only support tensorflow right now
3) Writing can be better

In Figure 1, make sure the names of components are consistent: either all start with nouns or verbs. The whole picture is not too illustrative.



Can switch the order of Figure 2 and Figure 3?

In Table 1, the description of ALL-CNN-C has a '?'. Is it intended?

Why not explain Table 2?

---

> ### Author Response · Authors · 2018-11-07
> **Response to the Comments of Reviewer 3**
>
> Dear Reviewer 3.
>
> thank you for your positive review. We are happy that you agree with us, that benchmarking stochastic optimization methods is a relevant project.
>
> We also want to address some of the points you have raised.
>
> Cons:
> 1) We agree, that it might take a large effort to convince others to use and contribute to DeepOBS. We designed DeepOBS to be as easy as possible to add new optimization methods. As long as you can implement your new optimizer in TensorFlow, you can add it to DeepOBS by sending us a pull request. We will invest time to run new optimization methods ourselfs, and, provided they give state-of-the-art performance, add them to the baselines.
> Additonally, benchmarking new optimization methods can take a lot of time, from setting up realistic test problems to computing fair baselines. With DeepOBS, this is unnecessary and researchers can spend more time on developing their optimization methods and less time on thinking about the benchmarking aspect. We hope that this is incentive enough to use DeepOBS.
> 2) We agree, that offering DeepOBS in other frameworks could be beneficial. However, we chose TensorFlow as it is arguably the most popular framework at the moment, and we had to start somewhere. We want to note that the actual software implementation is only a part of this paper.
> 3) If you can point us to some examples of bad writing in the paper, we would be very happy to address and re-write them and improve or clarify the sections.
>
> We also addressed the minor points you mentioned:
> We changed the names in Figure 1 to be more consistent. We hope that the picture is now more informative.
> In the current version, Figure 2 and 3 are switched now.
> We fixed the "?" in Table 1. It was the result of a typo in a citation. Thanks for noting this.
> We added an explanation for Table 2.
>
> Please note, that by making these changes the paper is now longer than 8 pages. We will work to reduce it to 8 pages again for the final version.

---

### Official Review · AnonReviewer2 · 2018-11-06
**An important _first_ step towards standardized procedures for benchmarking optimizers in deep learning.**

**Rating:** 6
**Confidence:** 4

**Review:**

This paper presents a new benchmark suite to compare optimizer on deep neural networks. It provides a pipeline to help streamlining the analysis of new optimizers which would favor easily reproducible results and fair comparisons.

Quality

The paper covers well the problems underlying the construction of such a benchmark, discussing the problems and models selection, runtime estimation, hyper-parameter selection and visualizations. It falls short however in some cases:

1. Hyper-parameter optimization
    While they mention the importance of hyper-parameter tuning for the benchmark, they leave it to the user to tune them without providing any standard procedure. Furthermore, they use grid search to build the baselines while this is known to be a poor optimizer [1].

2. Estimated runtime
    Runtime is estimated for a single set of hyper-parameters of the optimizer, but some optimizer may have similar or roughly similar results for a large set of hyper-parameters that widely affects the runtime. The effect of the hyper-parameters should be taken into account for this part of the benchmark.

3. Interpretation
    Such a benchmark should makes it easier for interpretation of results as the authors suggests. However, the paper does not convey much interpretation in section 4, beside the fact that results are not conclusive for any baseline. Results of the paper seem low, but they are difficult to verify since the plots are not very precise. For instance Wide ResNet-18-8 reports 1.54% test accuracy on SVHN [6] while this paper reports ~ 15% for the Wide ResNet 18-4 version. Figure 2 is a good attempt at making interpretations of sensitivity of optimizers' hyper-parameters but has limited interpretability compared to what can be found in the literature [2].

4. Problems
    There is an effort to provide varied types of problem, including classical optimization functions, image classification, image generation and language modeling. The number of problems consists mostly of image classification however and is very limited for image generation and language modeling.

Clarity

The paper is well written and easy to understand in general.

On a minor note, most figures are difficult to read. Side nodes on figure 1 does not divide clearly without any capital letter or punctuation at the end of sentence. Figure 2 should be self contained with its own legend. Figure 3 is useful for a visual impression of the speed of convergence but a histogram would be necessary for a better visual comparison of the different performances.

Section 2.2 has a confusing terminology for the "train valid set". Is it a standard validation set?

Originality

There is virtually no benchmarks for optimizers available for the community. I believe a standardized procedure for comparing optimizers can be viewed as an original contribution.

Significance

Reproducibility is a problem in machine learning [3, 4] and optimizers' efficiency on deep neural networks generalization performance is still not very well understood [5]. Therefore, there is a strong need for a benchmark for sound comparisons and to favor better reproducibility.

Conclusion

The benchmark presented in this paper would be an important contribution to the community but lacks a few important features in my opinion, in particular, sound hyper-parameter optimization procedure and sound interpretation tools. On a skeptical note, I doubt the benchmark will be used extensively if the results it provides yield no conclusive interpretation as reported for the baselines. As I feel there is more work needed to support the goals of the paper, I would suggest this paper for a workshop. Nevertheless, I would not be upset if it was accepted because of the importance of the subject and the originality of this work.

[1] Bergstra, James, and Yoshua Bengio. "Random search for hyper-parameter optimization." Journal of Machine Learning Research 13, no. Feb (2012): 281-305.
[2] Biedenkapp, Andre, Joshua Marben, Marius Lindauer and Frank Hutter. “CAVE : Configuration Assessment , Visualization and Evaluation.” In International Conference on Learning and Intelligent Optimization (2018).
[3] Lucic, Mario, Karol Kurach, Marcin Michalski, Sylvain Gelly, and Olivier Bousquet. “Are GANs Created Equal? A Large-Scale Study.” arXiv preprint arXiv:1711.10337 (2017).
[4] Melis, Gábor, Chris Dyer, and Phil Blunsom. “On the state of the art of evaluation in neural language models.” arXiv preprint arXiv:1707.05589 (2017).
[5] Wilson, Ashia C., Rebecca Roelofs, Mitchell Stern, Nati Srebro, and Benjamin Recht. "The marginal value of adaptive gradient methods in machine learning." In Advances in Neural Information Processing Systems, pp. 4148-4158. 2017.
[6] Sergey Zagoruyko and Nikos Komodakis. Wide residual networks. arXiv preprint arXiv:1605.07146, 2016

-----------
Revision
-----------

In light of the discussion with the authors, the revision made to chapter 4 and in particular the proposed modifications to section 2.4 for a camera-ready paper, I revise my score to 6.

---

> ### Author Response · Authors · 2018-11-07
> **Response to the Comments of Reviewer 2**
>
> Dear Reviewer 2,
>
> thank you very much for your constructive review.
> We are happy that you agree with us that a benchmarking suite would be an important step. While we acknowledge that the presented solution is not optimal, we would argue that it significantly improves on the current status quo. Just like Reviewer 1, we worry "that people will still find minor quibbles with particular choices or tasks in this suite, and therefore continue to use bespoke comparisons". We believe that the improvement of DeepOBS compared to the status quo (which often is to just use MNIST and CIFAR10 and compare to SGD or Adam) is larger than the step from DeepOBS to where we hope to be.
>
> We also want to address the shortcomings you mentioned in your review.
>
> 1. We believe that we can split this critique in two aspects. Firstly, the hyper-parameter optimization that we do for our baselines. While, we agree that grid search is not at all an optimal approach, we would argue that it is the method most common in practice (for example [1, 2]). The main goal of our baselines are to be a realistic comparison. We plan to include more sophisticated baselines in the future, for example ones that include learning rate decay schedules. We could tune these schedules with more complex methods than grid search, to provide a more challenging competition.
> The second part is that we don't provide a hyperparameter tuning method for the user. We did this on purpose. A hypothetical user of DeepOBS might want to highlight that their new optimization method gets good results using default hyperparameter values, while also showing that tuning those parameters a little bit can give you even better results. Therefore, we believe that the choice of hyperparameter tuning method should be left to the user. As long as they document this tuning process, and report the final hyperparameters on each test problem, the results are still comparable even when different tuning methods are used.
> 2. It is an interesting point you raised here. Indeed we only estimate the runtime for a single set of hyperparameters. However, the used hyperparameters for this estimation is flexible. In the scenario that you describe, the best option for the user of DeepOBS would be to do the estimation step twice for both settings and report both numbers.
> 3. We will indeed double-check the results of the Wide ResNet on SVHN. In contrast to the original paper, we do not use Nesterov momentum, nor a learning rate decay schedule. We also train for less epochs. The point of the test problems is not to provide state of the art results, but to compare the performances of optimization methods. Nevertheless, we will check our SVHN results and are currently running new experiments. Thanks for pointing this out.
> 4. While we agree that the set of test problems is a bit biased towards image classification, we also believe that this set is much more exhaustive than what is currently used in practice (which is often just MNIST and CIFAR10). If there is a specific test problem that you would like us to add, we would gladly do so. We see this set of test problems as a starting point and DeepOBS can be continuously improved and extended.
>
> We also tried to address the notes on clarity and changed the figures accordingly.
>
> In section 2.2 we mention a "train eval set", which is not a standard validation set. We use this train eval set, whenever we want to evaluate our training performance. We distinguish between using the training data to train, and using the training data to evaluate the performance on it. During this "training evaluation phase", we evaluate on the a set that is as large as the test set and also use the neural network in architecture in "evaluation mode" (for example we do not use dropout). This allows for a fairer comparison between test loss and train loss as both are computed in the same way.
>
>
> We hope that by addressing your points we were able to alleviate some of your concerns. You agree with us and the other reviewers that a benchmarking suite for deep learning optimizer would be a significant step and a useful tool for the field and that currently no such tool exists. We kindly ask you to reconsider your evaluation of the paper in light of this response.
>
>
> [1] Diederik Kingma, and Jimmy Ba. "Adam: A Method for Stochastic Optimziation" Proceedings of 3rd
> International Conference on Learning Representations (ICLR), 2015.
>
> [2] Tao Lin, Sebastian Stich, and Martin Jaggi. "Don't Use Large Mini-Batches, Use Local SGD" arxiv, 2018.

---

> > ### Comment · AnonReviewer2 · 2018-11-07
> > **Response to the comments of authors**
> >
> > Although I agree that the improvement of DeepOBS in its current state is better than the status quo, I do not believe this is a reason substantial enough to accept the current version of the paper as a conference paper.
> >
> > 1. Leaving it to the researcher to do the hyper-parameter search without standard procedures is dangerous. Researchers could easily report results overfitting the test set, although hopefully the use of different seeds could alleviate this problem. What is a realistic comparison should be defined more precisely. In my mind what realistic would mean is that the difficulty of the problems is similar to those tackled in research, that the complexity of the architectures are similar and that the computational budget is one available to most researchers. Bad practices like hyper-parameter optimisation on the test set or unequal hyper-parameter optimisation on different benchmarks should not be included as what is realistic, even though this is unfortunately something common in our community. My point on unequal optimization on different benchmarks is the reason why I do not think we can divide the critique in two aspects. Optimizers should be compared on an equal footing, which means the procedure used to compute the results of the baselines should be the same as the one used for the new optimizers of interest.
> >
> > 2. The proposed solution would not scale for many set of hyper-parameters. There should be a measure of runtime sensitivity across many different hyper-parameter values weighted by the corresponding generalization performances. This is also related to my criticism on the limited interpretability of Figure 2, in the sense that such runtime sensitivity could be measured using tools as those presented in [2].
> >
> > 3. I agree that state-of-the-art results is not necessarily a prerequisite for such a benchmark. However, if a given setup normally yields better result than what is reported as a baseline, how confident can we be that this baseline is reliable? Could it be that improvements on a new optimizer over the baselines could also be observed on SGD itself given a better hyper-parameter tuning? I would like to reiterate my criticism on interpretability here, which was not addressed in the comment. I strongly believe that the introduction of such a benchmark should be accompanied with improved methods of analysis otherwise the conclusions that one can make from the benchmark are likely to be brittle.
> >
> > 4. I agree that the variety of problems presented is a good first step, and would be sufficient as the first step.
> >
> > Thank you for the clarification about the "train eval set". I wonder why you do not use a validation set however.
> >
> > I truly appreciate the nature of your work and I sincerely hope my comments are encouraging you progressing further. Although I acknowledge the importance of its nature, I believe the lack of standardized hyper-parameter optimization procedures and analysis methods is a serious issue for such a benchmark.

---

> > > ### Author Response · Authors · 2018-11-08
> > > **Second Response**
> > >
> > > A general remark regarding all your comments. There is a very important trade-off between exhaustive benchmarking (all the information and plots that we want to see) and ease of use/computational cost. When designing DeepOBS we tried to not increase the (computational) effort for the researchers, while substantially increasing the quality of the benchmark.
> > >
> > > 1. We take your point about hyperparameter fitting. But let’s be clear: There is no widely accepted framework for the adaptation of hyperparameters such as step and batch sizes, etc. Which framework would you recommend we impose on our community to provide a level playing field for *all* ongoing research in deep learning optimization? Hypergradients, Learning to learn, Probabilistic Line searches, or Barzilei-Borwein? If we decided to pick any of these and force people to use it, would that convince you to accept this paper? And do you think it would increase the user base, or rather restrict it?
> > >
> > > 2. What you are describing is definitely something that is desirable. However, a runtime sensitivity would increase the effort to run such a benchmark drastically, especially in the case of many hyperparameters, which is why it is not common in deep learning optimization papers. One possible trade-off would be to estimate the runtime of the optimal hyperparameter setting of each test problem and report it in Table 2 along with the iterations as a relative factor. This would require at most 8 runtime estimations, while still providing some insights into what runtimes are to be expected in practice.
> > >
> > > 3. We are looking into our setup of this test problem to see why it produces worse results than reported in the paper. We don't believe that a drastically better results can be obtained by simply tuning the learning rate more. When looking at Figure 2, we would argue that it seems very unlikely that a drastically improved performance can be found from a learning rate, we didn't test. If sampling the hyperparameters even more (say on a loggrid with 100 points) would convince you, we can update our baselines accordingly.
> > >
> > > We realize now that calling it a "train eval *set*" might be confusing. It is not a separate data set (it is just in many ways implemented like one). What we call "train eval set" is just the evaluation of the training data set in the same fashion you would evaluate the test set (same size, not using dropout, etc.) This is a better estimator for training performance, than using the regular mini-batch train loss and train accuracy, we get while training. Since we want to assess training performance, it would not make sense to use a validation set.

---

> > > > ### Comment · AnonReviewer2 · 2018-11-15
> > > > **Response to the comments of authors**
> > > >
> > > > There seems to be a misconception that hyper-parameter optimization methods increase the computational cost. One of the reasons grid-search is not recommended is precisely because it is an inefficient way of doing hyper-parameter optimization, leading to wasted computations. Otherwise, I certainly agree that the exhaustiveness of the benchmark should be limited so that the computational cost is affordable to most researchers and that the usability should be simple enough to avoid scaring researchers away. I do not believe my comments are in contradiction with this. Our disagreement seems to be about how expensive the hyper-parameter optimization methods are and how simple or complex the runtime evaluation is.
> > > >
> > > > 1. There is no need to force researchers to use a specific hyper-parameter optimization method. What is needed is a clear procedure about how to optimize them so that baselines are comparable. That means, at least one set of baselines optimized with a specific framework defined by you, and clear guidelines about how baselines should be optimized if researchers want to use a different hyper-parameter optimization method. Although there is no widely accepted framework for the optimization of hyper-parameters, there is agreement that grid-search is the worst method. For the specific case with limited number of hyper-parameters such as step and batch sizes, simple Bayesian optimization methods have proved to work well.
> > > >
> > > > 2. A better tradeoff would be to reuse the available information about hyper-parameter optimization. Given that hyper-parameter optimization is being executed, we should already have plenty of results. The problem for runtime evaluation is that we need to run the baselines again so that all runtime evaluations are done on the same hardware. If the results of the baseline's hyper-parameter optimization was provided, the runtime evaluation sweep could be easily automated and given that each execution is fairly short (only two epochs), the total execution time would be less than a single training.
> > > >
> > > > 3. It would be surprising indeed that fine-tuning of the learning rate alone would have a dramatic effect. The main problem is probably the lack of learning rate decay schedule. As suggested in [1], it is better to compare SGD with adaptive gradient methods by using a learning rate decay schedule.
> > > >
> > > > I do understand the usefulness of the train eval set. To expand my previous comment, I am wondering why you do not also use a validation set because standard practice is to use it to select the best model (hyper-parameters) and then make final comparisons on the test set. What is done in the current work is both selections and final comparisons on the test set, which is not recommended.
> > > >
> > > > [1] Wilson, Ashia C., Rebecca Roelofs, Mitchell Stern, Nati Srebro, and Benjamin Recht. "The marginal value of adaptive gradient methods in machine learning." In Advances in Neural Information Processing Systems, pp. 4148-4158. 2017.

---

> > > > > ### Author Response · Authors · 2018-11-20
> > > > > **Third Response**
> > > > >
> > > > > Thanks for your continuing interest in the conversation! We answer to your points briefly below, but at this point we think it is important to return to the core premise of the paper and not get tied up in details. We understand that you would like to have the ideal solution. But at this point practice in the field is very far away from what you are asking for. We wanted to provide a benchmark tool that is practical, easy to use, and does not impose unrealistic or counterproductive constraints on researchers. We continue to believe that our work significantly improves the current standards in benchmarking for deep learning.
> > > > >
> > > > > Briefly on your points:
> > > > >
> > > > > 1. You asked for "one set of baselines optimized with a specific [hyperparameter optimization] framework defined by you". We are providing that, using the most basic framework possible: a (log) grid search with a fixed budget. You are arguing that this is a bad method and we wholeheartedly agree. But we are sure that you would agree that it might well be the most common method in practice. As you point out yourself, researchers are free to use any other hyperparameter optimization method, given that they re-tune the hyperparameters of the baseline methods (SGD, etc.) using the same method (we will add a paragraph to the paper to point this out more clearly). The baselines provided by us are purely for convenience and that's why we think it makes sense to keep the overhead needed to compare a new method to them as small as possible by choosing a simplistic tuning protocol.
> > > > >
> > > > > 2. We acknowledge that a runtime evaluation across hyper-parameter settings would be ideal (many things are ideal, but not all are feasible). We do not enforce this because for many methods (in particular first-order ones) runtime per step is independent of the hyper-parameters, and it seems silly to force people to show this over and over again. Note, though, that anyone can use the provided tools to provide exactly the sweep you suggested. We include a remark in the updated paper, suggesting to estimate runtime as a function of hyper-parameter settings where necessary.
> > > > >
> > > > > 3. We plan to include SGD (as well as Momentum and Adam) with learning rate decay schedules as a second, more challenging baseline in the future.

---

> > > > > > ### Comment · AnonReviewer2 · 2018-11-26
> > > > > > **Response to the comments of authors**
> > > > > >
> > > > > > I agree not to get tied up in details. The discussion about the runtime analysis got too much into details indeed.
> > > > > >
> > > > > > I do not agree that my criticisms are assimilable to an ideal solution. The goal of this framework is to provide "protocol[s] for the quantitative and reproducible evaluation of optimization strategies for deep learning". A strict minimum for this is that the protocols in question should be sound and provide insightful analyses. To constrain ourselves to current bad practices does not satisfy the strict minimum in my opinion, even though it is done for the sake of simplicity and usability.
> > > > > >
> > > > > > The importance of using exactly the same method as the one used for the benchmark to execute hyper-parameter search should be stressed out very clearly in the paper, explaining why and how to do it. This would indeed partly solve one of my main criticism for this paper.

---

> > > > > > > ### Author Response · Authors · 2018-11-28
> > > > > > > **Fourth Response**
> > > > > > >
> > > > > > > We do think that the protocol we suggest is sound and provides insightful analyses, even though we agree that we should strive for even better practices in the future.
> > > > > > >
> > > > > > > Regarding the hyperparameter tuning: During the last revision we added the following comment to Chapter 4 (which discusses the baseline results): "While we are using a log grid search, researchers are free to use any other hyperparameter tuning method, however this would require re-running the baselines as well."
> > > > > > >
> > > > > > > Unfortunately, the ICLR revision period has ended, but should this paper be accepted, we would like to put more emphasis on this comment in the camera-ready version. We would also like to add some clarifying comments to Section 2.4 (which discusses hyperparameter tuning) to
> > > > > > > - explain our choice for a log grid search as the baseline hyperparameter tuning procedure,
> > > > > > > - point out and cite other hyperparameter tuning methods,
> > > > > > > - and emphasize that using another hyperparameter tuning method requires re-running the baselines with that protocol.
> > > > > > >
> > > > > > > We thank you for your comments and suggestions and are looking forward to the final decision.

---

### Meta-Review · Area_Chair1 · 2018-12-10
**a useful benchmark for deep learning optimizers, but limited research contribution**

**Confidence:** 3
**Recommendation:** Accept (Poster)

**Metareview:**

The field of deep learning optimization suffers from a lack of standard benchmarks, and every paper reports results on a different set of models and architectures, likely with different protocols for tuning the baselines. This paper takes the useful step of providing a single benchmark suite for neural net optimizers.

The set of benchmarks seems well-designed, and covers the range of baselines with a variety of representative architectures. It seems like a useful contribution that will improve the rigor of neural net optimizer evaluation.

One reviewer had a long back-and-forth with the authors about whether to provide a standard protocol for hyperparameter tuning. I side with the authors on this one: it seems like a bad idea to force a one-size-fits-all protocol here.

As a lesser point, I'm a little concerned about the strength of some of the baselines. As reviewers point out, some of the baseline results are weaker than typical implementations of those methods. One explanation might be the lack of learning rate schedules, something that's critical to get reasonable performance on some of these tasks. I get that using a fixed learning rate simplifies the grid search protocol, but I'm worried it will hurt the baselines enough that effective learning rate schedules and normalization issues come to dominate the comparisons.

Still, the benchmark suite seems well constructed on the whole, and will probably be useful for evaluation of neural net optimizers. I recommend acceptance.